# Utilising Sentinel-1’s Orbital Stability for Efficient Pre-Processing of Radiometric Terrain Corrected Gamma Nought Backscatter

**DOI:** 10.3390/s23136072

**Published:** 2023-07-01

**Authors:** Claudio Navacchi, Senmao Cao, Bernhard Bauer-Marschallinger, Paul Snoeij, David Small, Wolfgang Wagner

**Affiliations:** 1Department of Geodesy and Geoinformation, TU Wien, 1040 Vienna, Austria; bernhard.bauer-marschallinger@geo.tuwien.ac.at (B.B.-M.); wolfgang.wagner@geo.tuwien.ac.at (W.W.); 2Earth Observation Data Centre for Water Resources Monitoring (EODC), 1030 Vienna, Austria; senmao.cao@eodc.eu; 3V.O.F. APSS, 4725 SJ Wouwse Plantage, The Netherlands; 4Department of Geography, University of Zurich, CH-8057 Zurich, Switzerland; david.small@geo.uzh.ch

**Keywords:** Sentinel-1, Synthetic Aperture Radar (SAR), Ground Range Detected (GRD), georeferencing, orbital tube, radiometric terrain correction (RTC), analysis-ready Data (ARD)

## Abstract

Radiometric Terrain Corrected (RTC) gamma nought backscatter, which was introduced around a decade ago, has evolved into the standard for analysis-ready Synthetic Aperture Radar (SAR) data. While working with RTC backscatter data is particularly advantageous over undulated terrain, it requires substantial computing resources given that the terrain flattening is more computationally demanding than simple orthorectification. The extra computation may become problematic when working with large SAR datasets such as the one provided by the Sentinel-1 mission. In this study, we examine existing Sentinel-1 RTC pre-processing workflows and assess ways to reduce processing and storage overheads by considering the satellite’s high orbital stability. By propagating Sentinel-1’s orbital deviations through the complete pre-processing chain, we show that the local contributing area and the shadow mask can be assumed to be static for each relative orbit. Providing them as a combined external static layer to the pre-processing workflow, and streamlining the transformations between ground and orbit geometry, reduces the overall processing times by half. We conducted our experiments with our in-house developed toolbox named *wizsard*, which allowed us to analyse various aspects of RTC, specifically run time performance, oversampling, and radiometric quality. Compared to the Sentinel Application Platform (SNAP) this implementation allowed speeding up processing by factors of 10–50. The findings of this study are not just relevant for Sentinel-1 but for all SAR missions with high spatio-temporal coverage and orbital stability.

## 1. Introduction

Synthetic Aperture Radar (SAR) missions have proven their great value for the monitoring of Earth system processes due to their all-weather and day and night operability [1]. The launch of the C-band SAR mission Sentinel-1A in 2014, followed by its companion Sentinel-1B in 2016, started a new era of SAR applications at high temporal and spatial resolution [2]. Combined dense backscatter time series from both satellites provide fertile ground for manifold applications such as surface soil moisture estimation [3], flood mapping [4], estimation of sea ice concentration [5], grassland mowing event detection [6], forest mapping [7], and snow melt monitoring [8]. Even though Near-Real-Time (NRT) operations were initially only foreseen over the oceans, there are now several Sentinel-1 NRT land monitoring services providing data products with a latency of only a few hours, including the Global Flood Monitoring Service (GFM) [9], the Copernicus Global Land Service (CGLS) [10] and the Integrated forest Fire Danger assessment System (IFDS) for the Austrian Alps [11,12,13].

The Sentinel-1 mission excels not only in its high spatio-temporal coverage and timeliness, with a six day repeat cycle at the equator and a maximum data latency of three hours [2], but also in terms of the quality of its backscatter images. However, many applications still rely on products whose radiometry is ellipsoid-corrected SAR backscatter—these show significant radiometric distortions over parts of the world that are not flat. Therefore, backscatter data acquired over mountainous and hilly sections are often discarded.

A milestone in overcoming this limitation was achieved by [14], who presented a method for computing a terrain flattened normalised radar backscatter coefficient, i.e., Radiometric Terrain Corrected (RTC) gamma nought $\gamma_{T}^{0}$. In recent years, $\gamma_{T}^{0}$ has been more and more widely used for improving the use of SAR data in undulated terrain [15,16] or across different orbits [17,18]. These achievements motivated the Committee on Earth Observation Satellites (CEOS) to select $\gamma_{T}^{0}$ as a standard for analysis-ready Data (ARD) [19].

The RTC operator to produce $\gamma_{T}^{0}$ has already been implemented in several SAR software packages [20,21,22], for example, in GAMMA [23], the InSAR Scientific Computing Environment version 3 (ISCE3) [24] and the Sentinel Application Platform (SNAP) [25]. Unfortunately, due to the complexity of the RTC algorithm, processing times are some orders larger than when doing basic orthorectification. Furthermore, several per-pixel metadata layers are required, including a shadow mask (“Data Mask Image”), the local contributing area (“Scattering Area Image”), and the local incidence angle (“Local Incident Angle Image”) (more details can be found in [26]). A remarkable step forward in terms of improving the quality and run time performance of radiometric terrain correction has been made by the novel area projection method presented in [27]. Still, when rolling out processing activities to more scenes, or even globally, one may be limited by compute and storage resources quickly. Consequently, multi-year $\gamma_{T}^{0}$ datasets are at the moment only available at continental scale, e.g., as presented in [28], who generated a CEOS compliant ARD normalised radar backscatter dataset over Africa.

Contributing to efforts for establishing worldwide ARD SAR data collections, we extend our recent work on utilising Sentinel-1’s orbital stability for efficient pre-processing of backscatter data [29]. For this, we use the Monte Carlo approach to ingest orbital fluctuations—which are in the order of around 50–60 m (radial RMSE)—into a RTC $\gamma_{T}^{0}$ pre-processing workflow. During the process, the impact of the orbital deviations on several ground-based layers is checked. Here, our main interest lies in the local contributing area used for terrain flattening ($A_{\gamma }$) and the shadow mask ($M_{S}$). When the impact of the orbital deviations on $A_{\gamma }$ and $M_{S}$ are very small or even negligible, then these two layers can be assumed to be static. This has already been proven in [29] for the (projected) local incidence angle ((P)LIA), with a very low standard deviation of only 0.005 degree. The major implication is that these layers do not need to be re-computed for each individual Sentinel-1 scene, and can be ingested to the workflow as a static input layer read from the storage.

A similar approach was pursued in the very recent study by [30], introducing a pixel-based gamma-to-sigma correction factor, which allows to go from Geometric Terrain Corrected (GTC) sigma nought $\sigma_{E}^{0}$ to RTC gamma nought $\gamma_{T}^{0}$ backscatter. By means of a parameter simulation using a number of scenes as input, the study revealed that the gamma-to-sigma layer behaves static over time and its variation falls well below Sentinel-1’s relative radiometric accuracy of 0.1 dB [31]. This ratio may be used to perform on-the-fly RTC computing on a stack of $\sigma_{E}^{0}$ images, resolving the need to perform radiometric terrain flattening in orbit geometry and thus significantly reducing computational requirements.

In this study, the focus is on generating directly $\gamma_{T}^{0}$ image stacks with a performant method, rather than relying on already available georeferenced backscatter data. Implementing the complete $\gamma_{T}^{0}$ pre-processing workflow in our Python package *wizsard* enabled us to analyse the impact of different variants of the pre-processing workflow on run time performance and radiometric quality. Replacing the repeated generation of selected pre-processing layers with their static representation—as long as they are satisfying the required radiometric accuracy level—will significantly improve run time performance. Thus, by taking the satellites’ high orbital stability into account, processing services will benefit from the insights provided in our study to produce a high-quality, normalised, dense backscatter time series within less arduous time frames.

## 2. Materials and Methods

### 2.1. Sentinel-1 $\gamma_{T}^{0}$ Pre-Processing Workflow

The Sentinel-1 $\gamma_{T}^{0}$ pre-processing workflow laid out in [14] requires more processing steps than the basic GTC $\sigma_{E}^{0}$ pre-processing workflow as presented in [32] or [29]. Instead of taking ellipsoid-based area values (provided in the Sentinel-1 metadata) for standardising the radar measurements to a certain backscatter convention, the actual illuminated area is estimated by integrating surface patches of a Digital Elevation Model (DEM) in the radar geometry. This area serves then as a standardisation factor to convert the radar brightness $\beta^{0}$ to radiometric terrain corrected gamma ($\gamma_{T}^{0}$) or sigma nought ($\sigma_{T}^{0}$). Finally, the RTC image may be reprojected to a standard map geometry to provide coregistered backscatter image stacks.

The RTC gamma nought workflow from [14] was graphically recycled in Figure 1 to create a baseline for our workflow modifications applied in Section 2.2. Similar to [29], we exclusively used Ground Range Detected (GRD), not Single-Look Complex (SLC) SAR as input data, and performed all georeferencing operations in the frame of the “LatLon” geographic projection system. The complete RTC gamma nought workflow is split into four groups:*Local parameterisation*: In the first step the ground-geometry-based layers are computed, i.e., the local contributing area, the shadow mask, and a Look-Up Table (LUT) containing azimuth and range indices. In this study, we use the shadow mask algorithm presented in [33]. By traversing the DEM from East to West or vice versa—depending on the orbit direction—the continuous analysis of the elevation angle allows to identify areas that are not visible to the sensor and thus do not contribute to the backscattered signal (occluded by shadow).*Radiometric adjustments*: The same method as described in [29], except that the calibration values refer to $\beta^{0}$ instead of $\sigma^{0}$.*Terrain flattening*: After bilinearly resampling the local contributing area (excluding pixels in shadow), into the orbit geometry, overlapping areas are cumulatively summed up. This area is then used to radiometrically normalise $\beta^{0}$ to $\gamma_{T}^{0}$, as explained in [14].*Georeferencing*: In the last step, $\gamma_{T}^{0}$ values are geocoded and resampled to the ground geometry at the desired pixel spacing of the final product.

Comparing this procedure with the GTC $\sigma_{E}^{0}$ workflow in [29] reveals two new elements that need to be investigated in terms of their variability over time: the local contributing area $A_{\gamma }$, and the shadow mask $M_{S}$. Their variability may be quantified by propagating Sentinel-1’s orbital distribution through the complete pre-processing chain, as previously carried out in [29]. If these layers can be declared static per relative orbit, the run time can be improved (on top of the improvements already gained by using a static (P)LIA layer in the $\sigma_{E}^{0}$ workflow).

A crucial part of defining the local contributing area as a static layer is the horizontal oversampling of the input DEM. The optimal (over-)sampling factor should ensure a good balance between radiometric quality and storage footprint, where [14] recommends an oversampling factor between 1 and 2. This should allow counteracting the introduced sampling artefacts due to the finer ground range resolution of the sensing geometry at backslopes.

In this respect, the authors of [27] have developed a novel method to achieve a much higher radiometric accuracy by reducing the magnitude of the required oversampling. Instead of only analysing each terrain facet locally, they propose to compute the normalisation area using a weighted mean for the whole facet in orbit geometry. This approach is very valuable for our study, since it allows reducing the order of oversampling for the static layers, also impacting the overall run time. With a focus on our representative study sites, a suitable value for the oversampling factor is defined in Section 2.4.

### 2.2. Sentinel-1 $\gamma_{T}^{0}$ Pre-Processing Workflow Enhancements

In the following, we present our modifications of the Sentinel-1 $\gamma_{T}^{0}$ pre-processing pipeline (Figure 1) by taking Sentinel-1’s unprecedented orbital stability into account. The performance of the new workflow was assessed by comparing it to SNAP and our own Python implementation of the Sentinel-1 $\gamma_{T}^{0}$ pre-processing workflow in *wizsard*. The SNAP workflow was realised with SNAP 8 (release details can be found here: [34]) by creating one Graph Processing Tool (GPT, [35]) graph comprising the following SNAP operators: (1) *Apply-Orbit-File* (2) *Calibration* (3) *Terrain-Flattening* (4) *Terrain-Correction* (5) *Subset* (6) *Write*. In addition to those already described in [29] for GTC sigma nought pre-processing, the *Terrain-Flattening* operator is added between the *Calibration* and *Terrain-Correction* operators to carry out the steps in Section 2.1. A single GPT graph file was used instead of one for each of the steps above to minimise intermediate data writing and reading. As a whole, it largely resembles the *wizsard* pre-processing chain. To allow a direct comparison with already existing run time results for $\sigma_{E}^{0}$ in [29], the latest major SNAP version 9 was not considered.

#### 2.2.1. Static $A_{\gamma }$ Layer

Motivated by previous results in [29] when discarding the (P)LIA computation for each scene yielded a significant performance improvement (and following Section 2.1), a static version of the local contributing area $A_{\gamma }$ and the shadow mask $M_{S}$ demonstrates a great potential of reducing processing effort. Assuming that both layers behave statically, they may be combined to one single static layer for each relative orbit by directly integrating the $M_{S}$ layer into the $A_{\gamma }$ layer as “no data”. The steadiness of both layers under Sentinel-1’s orbital variations will be investigated in more detail in Section 3 by means of Monte Carlo simulation. This combined layer may then be passed as an *a priori* input to the Sentinel-1 pre-processing chain, in addition to the DEM data.

Using our own Python implementation of the Sentinel-1 $\gamma_{T}^{0}$ pre-processing workflow in *wizsard* as a starting point, we decoupled the computation of the local contributing area and masking of shadow areas from the per-scene pre-processing pipeline. Consequently, the generation of the Look-up Table (LUT) is the only remaining operation in the *Local parameterisation* step.

#### 2.2.2. Single LUT

In many SAR pre-processing workflows Look-up Tables (LUTs) are generated twice, once during *Local parameterisation* and once during *Georeferencing*. This slows the workflow significantly. Particularly in SNAP, those two operators are completely separated, even when similar operations take place and the same intermediate data layers are utilised. This inefficiency can be avoided by keeping the first (oversampled) LUT in memory and providing it as direct input to the georeferencing routine. The azimuth and range indices may then be down-sampled to the target sampling of the $\gamma_{T}^{0}$ backscatter product by means of an average or Gaussian-weighted average. Note that this modification requires more RAM since the LUT resides in memory during the whole process.

#### 2.2.3. RTC Area Projection (RTC-AP)

In addition to efficiently managing transformations between the ground and orbit geometry with LUTs, radiometric enhancements may be achieved by using the novel RTC-AP algorithm, as presented in [27]. Instead of selecting bilinear resampling to project the local contributing area from ground to orbit, as initially proposed in [14], Shiroma et al. [27] recommend to warp the coordinate grid defined by the (oversampled) DEM to the scene geometry, and cumulatively sum it with the corresponding weighted $A_{\gamma }$ values.

The implementation of the area projection in *wizsard* follows the recipe described in [27], but does not consider optional performance improvements, as for instance to compute only once the weights for adjacent pixel edges. In our case, the local contributing area is not computed on the fly and is directly retrieved from the static $A_{\gamma }$ layer.

#### 2.2.4. Benchmarking Environment

To assess the performance of the altered Sentinel-1 $\gamma_{T}^{0}$ pre-processing chains, *wizsard* was repeatedly executed on a dedicated Linux machine with the same setup as used in [29], i.e., 4 cores and 32 GB RAM. Since the workflow modification in Section 2.2.2 keeps the LUT in memory and thus exceeds the initial RAM capacity setup, the available RAM was limited only for the basic version of Sentinel-1’s $\gamma_{T}^{0}$ pre-processing workflow.

### 2.3. Input Data

To analyse the deviations of the different pre-processing layers driven by the satellites’ orbital variations and to perform a run time comparison of different workflow setups, the exact same input data as described in [29] were considered (i.e., three Interferometric Wide (IW) swath, Ground Range Detected (GRD), and high-resolution (H) scenes from relative orbit 168 over Norway, Austria, and Benin; cf. Table 1 and Figure 1 in [29]). This enables comparability of orbital deviations and their propagated outcome, and allows benchmarking of run time results to existing ones measured at the $\sigma_{E}^{0}$ workflow.

In [29], we introduced a data-driven, semi-empirical model to represent average orbital trajectories and their deviations per relative orbit. The model relies on historical and precise orbit data (referred to as AUX_POEORB [36]) of the two-satellite constellation of Sentinel-1A and Sentinel-1B for the years 2017–2020. It samples positions (*p*) and velocities (*v*) at orbital reference points (${\hat{p}}_{\left\{x,y,z\right\}}$, ${\hat{v}}_{\left\{x,y,z\right\}}$) and their respective deviations formulated via discrete Probability Density Functions (PDF) of the orbital residuals ($\delta p_{\left\{x,y,z\right\}}$, $\delta v_{\left\{x,y,z\right\}}$), every 10 s in along-track direction. Also in this publication, we focus on the relative orbit number 168, which revealed orbital fluctuations in the order of 50–60 m on average for the whole revolution (cf. Figure 2 in [29]).

The only ground-based input layer to the pre-processing workflow is the 30 m Copernicus DEM [37], combined with geoid undulations retrieved from the Earth Gravitational Model (EGM) 2008 [38]. The Copernicus DEM given in ellipsoid heights was then regridded to the desired product pixel spacing of 10 m for each of the three scenes using cubic resampling.

### 2.4. Oversampling Analysis

As introduced in Section 2.1, an important processing parameter is the oversampling factor that governs the spatial sampling of the DEM used during the local parameterisation, i.e., the computation of the LUT, the shadow mask, and the local contributing area. In theory, a proxy of the required oversampling factor can be calculated by considering the Nyquist-Shannon sampling theorem [39]. For this, the local terrain conditions need to be considered, i.e., the terrain slope $\beta_{T}$ in range direction, PLIA defined on the ellipsoid $\theta_{\mathrm{PLIA},E}$, PLIA referring to the terrain point $\theta_{\mathrm{PLIA},\;T}$, and the desired ground range sampling $\Delta r_{\mathrm{GR},\;E}=10\;m$. With the latter and $\theta_{\mathrm{PLIA},\;E}$, it is possible to obtain the equivalent slant range $\Delta r_{\mathrm{SR},\;E}$ (Equation (Equation 1)):(1)$$\Delta r_{\mathrm{SR},\;E}=\Delta r_{\mathrm{GR},\;E}\;\sin\left(\theta_{\mathrm{PLIA},\;E}\right)$$

The local ground range sampling $\Delta r_{\mathrm{GT},\;T}$ is then found with Equation (Equation 2), which is a slightly modified version of Equation (10.4) in [40].
(2)$$\Delta r_{\mathrm{GR},\;T}=\frac{\Delta r_{\mathrm{SR},\;E}}{\sin(\theta_{\mathrm{PLIA},\;T}-\beta_{T})}$$

Since $\Delta r_{\mathrm{GR},T}$ refers to the viewing direction of the sensor and not to the North (Y) and East (X) component of the DEM’s coordinate system, Equation (Equation 3) projects $\Delta r_{\mathrm{GR},T}$ onto the coordinate axes by taking into account the local azimuth of the satellite $\alpha_{T}$.
(3)$$\left[\begin{array}{c} \Delta r_{X,\;T}\\ \Delta r_{Y,\;T} \end{array}\right]=\Delta r_{\mathrm{GR},\;T}\;\left[\begin{array}{c} \cos(\alpha_{T})\\ \sin(\alpha_{T}) \end{array}\right]$$

Setting the maximum component in relation to the given (equal) spatial sampling of the DEM Δs=10m, i.e., the target sampling of our product, and applying the sampling theorem, allows us to estimate the required oversampling factor $s_{o}$.
(4)$$s_{o}=2\;\frac{\Delta s}{\max(\left\{\Delta r_{X,\;T},\Delta r_{Y,\;T}\right\})}$$

This quantity is mapped in Figure 2 for each of the three regions of interest in Norway, Austria, and Benin. Foreslopes are masked to visually stress the importance of the required oversampling at backslopes. Two points can be seen clearly: the steeper the terrain facing away from the sensor and the closer in near-range, the larger the required oversampling factor. In extreme cases, $s_{o}$ needs to be around 4 to guarantee a more or less artefact-free terrain flattening procedure. Considering an oversampling factor of 4, the input DEM would contain 16 times more pixels to store, load, and iterate over to compute the local contributing area, which is a thorn in the side of an efficient pre-processing workflow. Fortunately, Shiroma et al. [27] presented a methodology to achieve radiometric quality with no or minimal oversampling by considering the actual area covered by the terrain facet in orbit geometry (instead of applying bilinear oversampling). The Appendix A offers further insights on aliasing effects in the accumulated normalisation area in orbit geometry for RTC gamma nought.

## 3. Uncertainty Propagation

Knowledge of the distribution of a system’s output can be obtained by feeding it with uncertainties assigned to its explanatory variables. There are several methods that allow one to propagate these uncertainties through the system in question, and impose certain statistical requirements on the distribution of the input parameters, for example, being a Gaussian. In this regard, Monte Carlo simulation offers the most flexibility and does not limit the complexity of the system—at the cost of computational effort [42]. Since the distributions of our orbital reference points are non-Gaussian [29], and the pre-processing workflow has some non-closed-form components, Monte Carlo simulation appears as an appropriate tool for the uncertainty propagation.

The uncertainty is expressed by the discrete PDFs of the orbital residuals $\delta p_{\left\{x,y,z\right\}}$ and $\delta v_{\left\{x,y,z\right\}}$ located at the orbital reference points ${\hat{p}}_{\left\{x,y,z\right\}}$ and ${\hat{v}}_{\left\{x,y,z\right\}}$. Sampling an orbital state vector from these PDFs along with providing the DEM as input allows one to execute the pre-processing workflow described in Section 2.1 to generate all layers. The horizontal and vertical accuracy of the Copernicus DEM was not considered in the uncertainty propagation to highlight only those deviations originating from orbital fluctuations.

Repeating the sampling *n* times provides an estimate of the variability of the ground-based layers of interest, i.e., the local contributing area $A_{\gamma }$ and the shadow mask $M_{S}$. Similar to [29], n=1000 yielded a stable description of the distribution by investigating the asymptotic behaviour of its sample mean.

### 3.1. Monte Carlo Simulations of $A_{\gamma }$ and $M_{S}$

The local contributing area $A_{\gamma }$ is the main output of the *Local parameterisation* step as described in Section 2.1. Repetitively computing $A_{\gamma }$ for different samples—drawn from the orbital distributions by means of Monte Carlo simulation—yielded the standard deviation $\sigma_{A_{\gamma }}$ as depicted in Figure 3. Overall, the maximum of $\sigma_{A_{\gamma }}$ was around 0.006 m${}^{2}$, which is a very small number compared to the size of the illuminated area.

Several interesting aspects can be observed in Figure 3: First, a scalloping pattern emerged along the azimuth direction in a very similar manner to the standard deviation of (P)LIA in [29]. This pattern was mainly caused by the actual definition of the orbital uncertainties, i.e., that they are only applied at the orbital reference points which are 10 s apart in along-track direction. Across-track, $\sigma_{A_{\gamma }}$ seemed to increase from near to far range, which can be explained by the enlargement of the projected illuminated area observed by the sensor.

Second, when comparing the three scenes, it appears that $\sigma_{A_{\gamma }}$ was lowest in Norway, increasing from North to South. This behaviour is bound to the oversampling of the “LatLon” coordinate system that features from the equator to the poles increasingly narrower distances between its grid points. The comparable magnitude of $\sigma_{A_{\gamma }}$ values in Austria and Benin might stem from the superposition of the *x* and *y* components of the orbit trajectory, which form a combined peak at mid-latitudes in Europe, as explained in Section 4.3 in [29].

Third, the zoom-in views provide a glimpse of the fine-scale, terrain-based variations of $\sigma_{A_{\gamma }}$, revealing larger values at slopes facing away from the sensor. This is caused by the widening of the projected illuminated area in the same manner as from near to far range.

In Figure 3, the standard deviation of $A_{\gamma }$ is overlaid with a shadow mask probability, which was computed by counting how often a pixel was classified as shadow and setting this quantity in relation to the number of simulations *n*. The first zoom-in view in Norway and Austria shows several regions being in shadow all the time, whereas in the second zoom-in view one is able to spot a few pixels with a probability of less than 100%. Thus, we can conclude that Sentinel-1’s orbital deviations do not significantly impact $M_{S}$—only at the edge of shadow areas, a very small number of pixels are not always in shadow for the same relative orbit.

### 3.2. Static Layer Realisation of $A_{\gamma }$ and $M_{S}$

The results of the simulation demonstrate that these parameters are appropriate as static layers and to be provided as external layers to a pre-processing workflow. As already introduced in Section 2.1, both $A_{\gamma }$ and $M_{S}$ can be merged by declaring pixels in shadow to be “no data” values. Such a combined static layer can be established conveniently per relative orbit. This would drastically reduce the required overall processing power and storage footprint compared to generating and storing these datasets for each observation.

However, to effectively benefit from the combined static layer, we would need to ensure that the small standard deviation of $A_{\gamma }$’s distribution ($\max\left(\sigma_{A_{\gamma }}\right)∼0.006\;m^{2}$) does not cause backscatter variations exceeding the relative radiometric accuracy of Sentinel-1’s C-band sensor, i.e., 0.1 dB (three sigma) [31]. The connection between $A_{\gamma }$ and $\gamma_{T}^{0}$ is formulated in Equations (24)–(26) in [14] and summarised in Equation (Equation 5).
(5)$$\gamma_{T}^{0}=K_{\gamma }\;\frac{\beta^{0}\;A_{\beta }}{A_{\gamma }}$$

This strict relationship allows to apply Gaussian error propagation to retrieve the standard deviation of $\gamma_{T}^{0}$’s distribution. By using the first derivative of Equation (Equation 5) as input to the error propagation formula, Equation (Equation 6) yields our estimated effect on the SAR radiometry, $\sigma_{\gamma_{T}^{0}}$:(6)$$\sigma_{\gamma_{T}^{0}}^{2}={\left(\frac{\partial \gamma_{T}^{0}}{\partial A_{\gamma }}\right)}^{2}\;\sigma_{A_{\gamma }}^{2}={\left(-K_{\gamma }\;\frac{\beta^{0}\;A_{\beta }}{A_{\gamma }^{2}}\right)}^{2}\;\sigma_{A_{\gamma }}^{2}$$

Figure 4 displays the result of the error propagation for each pixel of the Austrian scene, represented as an Inter-Decile Range (IDR), i.e., as the difference between the 90th and 10th percentile. This metric was chosen to visualise the spread of $\gamma_{T}^{0}$’s distribution in dB, and not in linear units. Overall, the IDR values were low, with a maximum of about ∼0.03 dB. This shows that $A_{\gamma }$ can be treated as static per relative orbit, in respect to Sentinel-1’s relative radiometric accuracy. The largest discrepancies occurred at backslopes, located in or close to shadow areas, confirming the need that pixels on the “ragged edge of shadow” should be masked [14].

In Figure 5 we present how a static layer might look for each of the three scenes. $A_{\gamma }$ was produced by considering the average orbit trajectory defined by the orbital reference points ${\hat{p}}_{\left\{x,y,z\right\}}$ and ${\hat{v}}_{\left\{x,y,z\right\}}$ and is overlaid with $M_{S}$ in orange.

The pixels of the static layer representing regions in radar shadow were not identified using the average orbital trajectory, but were rather derived from the simulated radar shadow mask probability. Giving preference to the best radiometric quality and taking note of the fact that pixels on the “ragged edge of shadow” are recommended to be masked [14], pixels with a shadow mask probability larger than zero were declared as shadow.

## 4. Results

The insights gained from the analysis on the impact of Sentinel-1’s orbital fluctuations on several pre-processing layers in Section 3 can now be combined with the proposed modifications of the Sentinel-1 $\gamma_{T}^{0}$ pre-processing chain presented in Section 2.2. In this section, we benchmark these workflows and run them repetitively on the dedicated project machine to identify the most performant setup for $\gamma_{T}^{0}$ generation. Finally, we display the difference in backscatter between the method performing best—based on radiometric quality and run time—and the original output of the basic pre-processing pipeline in Section 4.3.

### 4.1. Run Time Benchmarking

To reliably assess the run time behaviour, we executed 50 times each pre-processing scenario presented in Section 2.2. Table 1 summarises the average values of the run time experiments and regroups the steps depicted in Figure 1:*Scene preparation*: Merges all scene- and orbit-related steps, including reading Level-1 data.*Auxiliary data preparation*: Comprises loading and preparation of all auxiliary layers, i.e., DEM data, and optionally, the static layer per relative orbit.*RTC*: Performs all steps under *Local parameterisation*, *Terrain flattening*, and *Georeferencing*, excluding I/O as indicated in Figure 1.*Data export*: Single step writing all encoded data as GeoTIFF files to disk. Since processing $\gamma_{T}^{0}$ consumes more RAM, we encoded backscatter data as scaled dB values and selected *Int16* as a data type.

The following abbreviated workflow names refer to the aforementioned variations of Sentinel-1’s $\gamma_{T}^{0}$ pre-processing chain: The original pre-processing setup once implemented in SNAP 8 (“SNAP 8”) and once in Python (“wizsard (base)”), the latter workflow without the on-the-fly computation of $A_{\gamma }$ and $M_{S}$ (“wizsard (static $A_{\gamma }$)”), the utilisation of a single LUT (“wizsard (single LUT)”), and finally significantly improving the radiometric quality with the RTC-AP method (“wizsard (RTC-AP)”). Only the total run time is shown for “SNAP 8”, since using a single GPT graph file does not allow the retrieval of run times in compliance with the listing above. The best benchmarking result of the Sentinel-1 $\sigma_{E}^{0}$ pre-processing workflow, shown in our previous study in [29], is also appended to the table to demonstrate the computational overhead when performing radiometric terrain correction.

Each implementation was executed with two oversampling factors, i.e., 1 and 2, to assess how the algorithms scale with a higher sampling of the input DEM data. An analysis on the radiometric and computational influence of oversampling has already been performed extensively in [27]; therefore, it is not considered necessary to further investigate here. Unfortunately, the limited and predefined set of resources did not allow SNAP 8 to be executed for oversampling factors larger than 1.

If we take a look at Table 1 and compare it with our findings in [29], similar conclusions can be drawn. Across all scenes, *Scene preparation*’s run time remained nearly constant due to a similar file size of the scenes. Furthermore, the projection system of the input DEM, i.e., the “LatLon” system, introduced an increasing oversampling with latitude. This caused a significant increase in run time from South to North for all ground-based processing categories, i.e., *Auxiliary data preparation*, *RTC*, and *Data export*.

Most remarkably, “wizsard (base)” is more than 10–50 times faster than “SNAP 8” in total run time. This performance boost might stem from the utilisation of *Numba* that achieves C-like speeds, and from the streamlining of similar computations. On the contrary, SNAP 8 has a very modular workflow setup allowing one to work intuitively with different satellite missions and SAR operators. It separates the *Terrain-Flattening* from the *Terrain-Correction* operator, and thus the performance is lower. We also observed issues during parallelisation (which might have been improved in more recent SNAP versions).

The first variation of the pre-processing chain replaces the dynamic computation of $A_{\gamma }$ and $M_{S}$ with the respective static layer. Compared to “wizsard (base)”, this scenario consumes ∼10–20% less time. Notably, this is less than the improvement with the static (P)LIA layer in [29], because $A_{\gamma }$ and $M_{S}$ are initially not written to disk. Reading the static $A_{\gamma }$ layer in addition to the DEM data takes a few seconds longer, where the difference scales with oversampling, but on the other hand, fewer computations in the *RTC* step reduce the run time significantly, e.g., by about 100 s for the scene located in Norway (not oversampled).

Introducing a single LUT that stores range and azimuth indices into the pre-processing workflow had the largest impact on the run time. This version of the workflow reduced run times by ∼10–20% on top of the improvement from the static $A_{\gamma }$ layer, leading to an overall boost of ∼30–40%. However, observing the RAM utilisation during processing, naively stored LUT values cost two to three times more RAM. This most probably exceeds common office computer capacities, but could still be an option for a high-performance computing (HPC) environment. Another strategy would be to improve *wizsard*’s RAM management.

Doubling the oversampling slowed down the reading of the input DEM data and static layer (*Auxiliary data preparation*) and the radiometric terrain correction (*RTC*) by two to three times. This order illustrates the general heavy workload when generating radiometrically adequate $\gamma_{T}^{0}$ backscatter, since Figure 2 indicates that we need at least an oversampling factor of around 4 when following the initial procedure in [14]. According to the validation in [27], and as visually emphasised in our Appendix A, the RTC-AP method provides the necessary framework for preserving a high radiometric quality. In this regard, RTC-AP without oversampling performs as well as bilinearly resampling $A_{\gamma }$ based on an oversampling factor greater than 5 to 7. In terms of run time statistics, Table 1 indicates that RTC-AP decelerates processing around two to three times, which is not in alignment with the extremely good performance of the algorithm as presented in [27]. *wizsard* only implements a very rudimentary version of the RTC-AP algorithm without streamlining certain parts, such as parallelised computations, or the duplicate processing of adjacent pixel edges. Yet, even under these circumstances, “wizsard (RTC-AP)” without oversampling is still close to “wizsard (single LUT)” with an oversampling factor of two.

### 4.2. Recommended Sentinel-1 $\gamma_{T}^{0}$ Pre-Processing Workflow

Based on all the insights gained so far, we selected the most performant components and updated the initial pre-processing workflow in Figure 1. The improved version of the workflow follows the “wizsard (RTC-AP)” setup and is shown in Figure 6. Instead of running computationally expensive vector operations to calculate $A_{\gamma }$ for each scene, a static $A_{\gamma }$ layer serves as an additional input parameter. Furthermore, the RTC-AP method justifies the usage of an input DEM and $A_{\gamma }$ that match the target sampling, since it mitigates sampling artefacts without applying oversampling. The updated processor may also take advantage of extensive RAM resources by storing information on the connection between orbit and ground geometry in a look-up table.

### 4.3. Backscatter Benchmarking

As a final experiment, we investigated how the proposed improvements (with Area Projection (AP) and a Static $A_{\gamma }$ Layer (SL)) affected georeferenced RTC gamma nought backscatter values $\gamma_{T-\mathrm{AP}-\mathrm{SL}}^{0}$. Figure 7 visualises the differences in backscatter between $\gamma_{T-\mathrm{AP}-\mathrm{SL}}^{0}$ and $\gamma_{T}^{0}$ that stem from the two workflow versions shown in Figure 6 and Figure 1, respectively. To focus on the positive impact of the RTC-AP method, both backscatter images were generated without oversampling. Shiroma et al. [27] have already demonstrated that higher oversampling factors do not significantly alter the level of $\gamma_{T}^{0}$ backscatter, especially for VV polarisation.

Figure 7 shows that the difference is very low in flat regions and tends to increase with steepness of the terrain. On the contrary, in undulated areas affected by layover and shadowing, the backscatter differences exceed Sentinel-1’s relative radiometric accuracy of 0.1 dB. The major portion of the differences can be attributed to the positive influence of the RTC-AP method featuring lower backscatter values on backslopes, where (under-)sampling artefacts are mitigated, and higher values on foreslopes. This radiometric improvement underlines the merit of using the RTC-AP method, allowing to exploit the full potential of RTC gamma nought backscatter in undulated terrain.

## 5. Discussion

Our examination on the impact of Sentinel-1’s orbital fluctuations on the local contributing area and shadow mask has clearly shown that these both can be declared per relative orbit as a static layer. With orbital deviations of around 50–60 m (RMSE), $A_{\gamma }$’s standard deviation $\sigma_{A_{\gamma }}$ reached a maximum of ∼$0.006\;m^{2}$. Propagating this quantity further to the backscatter level resulted in an inter-decile range of ∼0.03dB in areas in or close to radar shadow. This is within the relative radiometric accuracy of Sentinel-1’s C-band sensor and agrees well with the variation of the gamma-to-sigma layer presented by [30].

A static $A_{\gamma }$ layer does not only boost the pre-processing run time, but is also highly beneficial to other applications involving geocoding and radiometric adjustments. For example, it can enable refinements of the radar geometry [14], or serve as input to the creation of RTC gamma nought composites that combine images from different orbits [17].

The same applies for a static $M_{S}$ layer, which is not only an integral part of $A_{\gamma }$, but is also an asset for many bio-geophysical parameter retrievals in mountainous terrain to mask unreliable backscatter values, e.g., [8,30]. Moreover, the extent of shadow areas appears to remain stable over time, as indicated by an almost constant shadow mask probability of 100%, as only a few pixels on the “ragged edge of a shadow” area were not classified as shadow in every simulation. As an outcome of our Monte Carlo analysis, we generated an $A_{\gamma }$ layer from the average orbital trajectory and masked pixels with a shadow mask probability larger than 0%. Instead of applying an empirically based buffer of 150–200 m on a layover-shadow mask layer, as demonstrated in [30], we recommend using the static shadow mask, applied per relative orbit. This should guarantee that information on backscatter values is not unnecessarily discarded, even in spatial vicinity to areas occluded by shadow. If one still needs to get rid of pixels close to radar shadows, one may rather use the IDR layer based on $\sigma_{A_{\gamma }}$ in combination with the relative radiometric accuracy of Sentinel-1’s C-band sensor.

To mitigate resampling artefacts emerging from transformations between the ground and orbit geometry—which are mainly located on backslopes and reduce the radiometric quality of the georeferenced $\gamma_{T}^{0}$ product—the DEM and the static $A_{\gamma }$ layer must be oversampled. According to our analysis in Section 2.3 and the aliasing effects visualised in Figure 7, an oversampling factor of around 4 or higher would be necessary to mitigate these artefacts as best as possible. However, whether the storage footprint of these external layers or on-the-fly resampling would quickly exceed system resources or dramatically increase processing times. Fortunately, with the novel RTC-AP algorithm from [27], it is possible to resolve the need for extreme oversampling by projecting the actual local terrain facet into the orbit geometry and sticking to the native resolution of the final $\gamma_{T}^{0}$ backscatter product—without applying DEM oversampling at all.

To evaluate and select the most performant Sentinel-1 $\gamma_{T}^{0}$ pre-processing workflow, we introduced different processing scenarios in Section 2.2 and benchmarked them in Section 4. Streamlining the basic workflow (shown in Figure 1) in *wizsard* reduced the run time by around 10–50 times compared to our external workflow reference implemented in SNAP 8. Additional improvements were achieved by removing the individualised per-scene computation of the local contributing area $A_{\gamma }$ and the shadow mask $M_{S}$ from the pre-processing chain and replacing it with a static version being ingested into the pipeline together with the DEM (∼10–20%).

A permanent LUT had the most notable impact on the run time, in the order of 30–40%—at a cost of an increased RAM utilisation. Doubling the oversampling factor behaved slightly more than one-to-one with a lower performance. Our rudimentary implementation of the RTC-AP method turns out to be even slower, but it is able to preserve adequate radiometric quality, which could only be approximated by using higher oversampling factors.

Together with the RTC-AP method, the “wizsard (RTC-AP)” workflow, as visualised in Figure 6, is our recommended way for Sentinel-1 $\gamma_{T}^{0}$ pre-processing. Despite the run time degradation due to the non-optimised area projection in the RTC-AP step, it is possible to achieve a performance similar to the most performant setup “wizsard (LUT)” with the next larger oversampling factor. Introducing parallelisation in the RTC-AP step together with the recommended optimisations by [27] might then perform well on both ends—minimising run times while preserving adequate radiometric quality.

The fastest $\gamma_{T}^{0}$ pre-processing pipeline is not able to compete with the $\sigma_{E}^{0}$ pre-processing chain, which has approximately 20% less run time. The weaker performance is caused by the need to traverse the DEM (or the LUT table) two times, once when integrating the local contributing area, and once when georeferencing the radiometric terrain corrected backscatter image. Thus, the clever approach of [30] to perform radiometric terrain correction on the fly—by applying a static gamma-to-sigma ratio image to a $\sigma_{E}^{0}$ datacube—is a big step forward to reduce processing costs for large-scale $\gamma_{T}^{0}$ production.

The procedure of simulating static layers per relative orbit as presented in both of our studies is extremely useful for generating all CEOS-ARD layers, without the need of averaging scene-related images over time, as suggested, e.g., in [30]. The SNAP-based solution presented in [30] still has some downsides in terms of processing performance, oversampling, and systematic artefacts. Together with our findings on the stable behaviour of (P)LIA in our previous paper [29], disk volume can be saved considerably when certain per-pixel metadata layers were declared static. Yet, there is still the challenge to distribute these layers to the end users, i.e., if they are attached to each scene, or provided via a different endpoint that offers them for a certain region of interest. Especially with other SAR satellite missions to come, a flexible approach for accessing and generating static layers is needed, where our framework could assist in producing these layers locally. To speed up pre-processing, we hope that open-source toolboxes like SNAP or ISCE3 take the insights from [27,30] and our studies into consideration, and will be designed to accept external static layers as input.

In our study, we focused only on a single relative orbit (168), but identified great potential for applying our methods to other Sentinel-1 orbits, or even other satellites. Operational and planned SAR missions like the RADARSAT Constellation Mission (RCM) [43], Sentinel-1C/D [44], NISAR [45], and ROSE-L [46], also operate in a small orbital tube with a radius of a few 100 m and thus could also benefit from the insights presented in our two publications for GTC sigma nought and RTC gamma nought. However, when orbital tubes are larger than this order of magnitude, our approach would reach its limits quickly. This has already been demonstrated by [30], where ALOS-1’s baseline variations of around 6500 m surpassed the acceptable range of radiometric variations.

Finally, the overall idea of statistically describing orbital trajectories and propagating their deviations through a geocoding process is valuable for many scenarios. For instance, when one wants to assess the impact of orbit maneuvers, apply sensor fusion, as, e.g., in [17], or create (normalised) backscatter models for sensor calibration and design [47]. It can help also quantifying the sensitivity of certain parameters on orbital variations in a radiative transfer or bio-geophysical model, and thus support applications such as flood mapping [4], wet snow mapping [8], surface soil moisture retrieval [48], and sea ice analysis [49].

## 6. Conclusions

Recent efforts to utilise Sentinel-1’s orbital stability for efficient pre-processing of GTC sigma nought backscatter $\sigma_{E}^{0}$ [29] were extended by analysing the state-of-the-art RTC gamma nought $\gamma_{T}^{0}$ workflow and its intermediate layers in a similar manner. This study clearly demonstrated the additional benefit of a static local contributing area $A_{\gamma }$ combined with a static shadow mask $M_{S}$ per relative orbit. Our proposed enhancements mitigate the overhead of computing and storing these intermediate layers per scene and forming orbit-ground geometry look-up tables twice, which allowed to reduce the overall run time by approximately half. This paves the way for generating large-scale or global $\gamma_{T}^{0}$ products in a reasonable time. With multiple SAR satellites and SAR-based services on the horizon, it is essential that data providers are prepared to supply the earth observation community with efficiently processed ARD backscatter data.

## Figures and Tables

**Figure 1 sensors-23-06072-f001:**
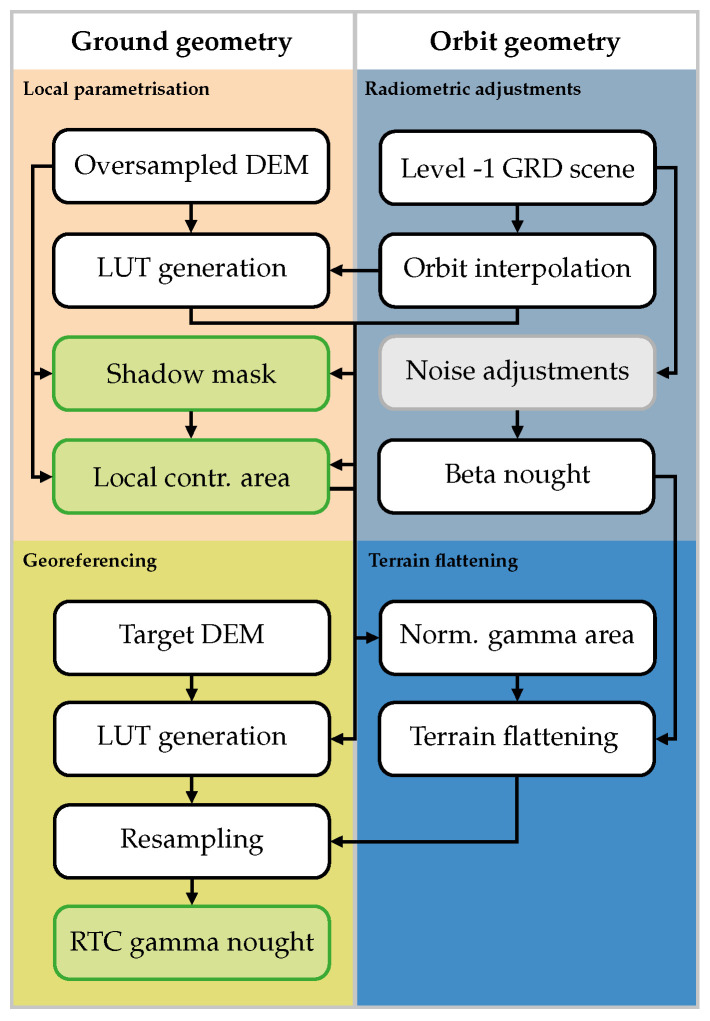
Sentinel-1 RTC gamma nought pre-processing workflow. Important intermediate layers are coloured in green and greyish boxes indicate common, but not mandatory steps.

**Figure 2 sensors-23-06072-f002:**
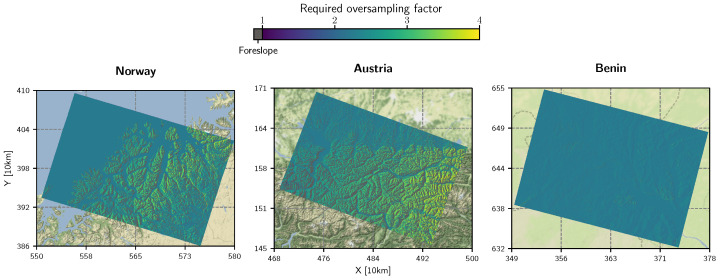
Required DEM oversampling factor shown for the three scenes introduced in [29]. Fore-slopes are masked in grey. To make the maps comparable across the large range in latitude, the Equi7Grid projection [41] was chosen in this and the following figures. Additionally, all data are shown on top of Stamen’s terrain-background map (map tiles by Stamen Design, under CC BY 3.0. Data by OpenStreetMap, under ODbL).

**Figure 3 sensors-23-06072-f003:**
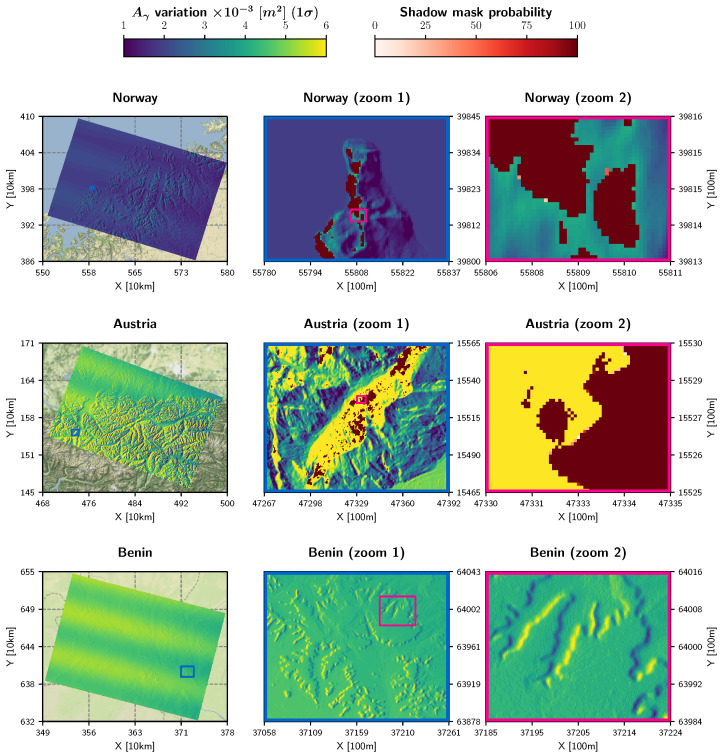
Local contributing area $A_{\gamma }$ sample standard deviation of the probability density functions resulting from the Monte Carlo simulations applied to the three scenes introduced in [29]. The shadow mask ($M_{S}$) probability is shown as an overlay. The coloured boxes indicate the extent of the zoom-in views on the right, i.e., navy-blue corresponds to “zoom 1” and pink to “zoom 2”.

**Figure 4 sensors-23-06072-f004:**
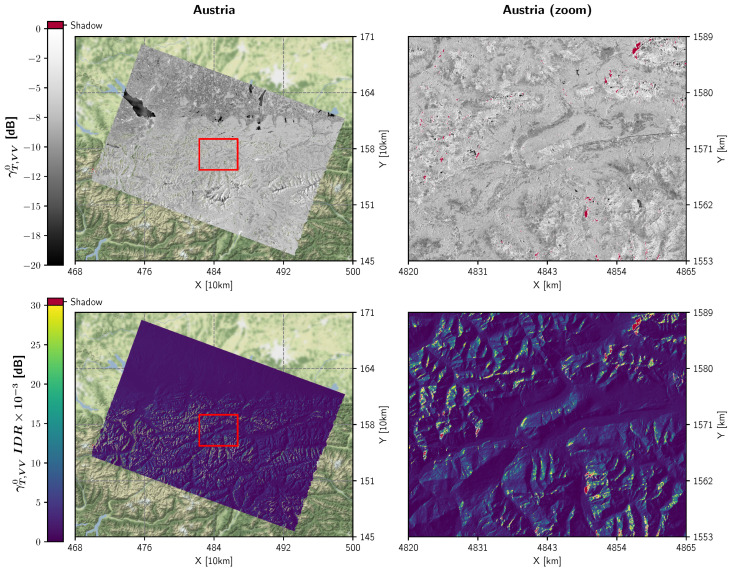
RTC gamma nought backscatter image of the Austrian scene, overlaid with a shadow mask (top). Inter-Decile Range (IDR) of the $\gamma_{T}^{0}$ backscatter distribution as function of $A_{\gamma }$ (bottom). The red boxes on the left side indicate the extent of the zoom-in view on the right side.

**Figure 5 sensors-23-06072-f005:**
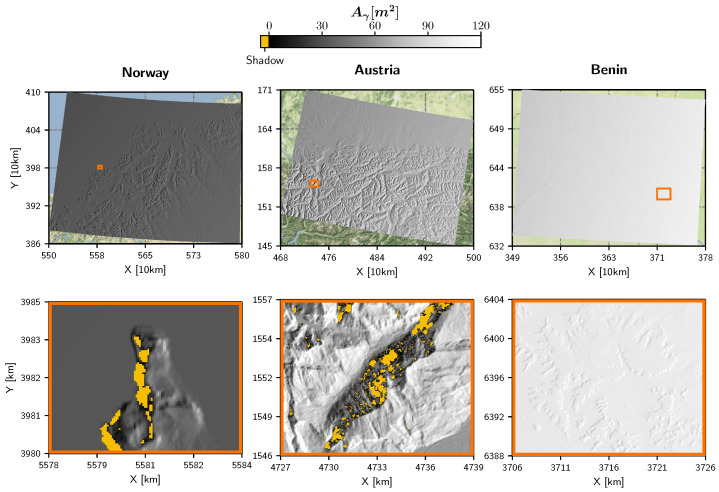
Static layer of the local contributing area $A_{\gamma }$ overlaid with the shadow mask $M_{S}$ and visualised for the three scenes introduced in [29]. The orange boxes indicate the extent of the zoom-in view at the bottom.

**Figure 6 sensors-23-06072-f006:**
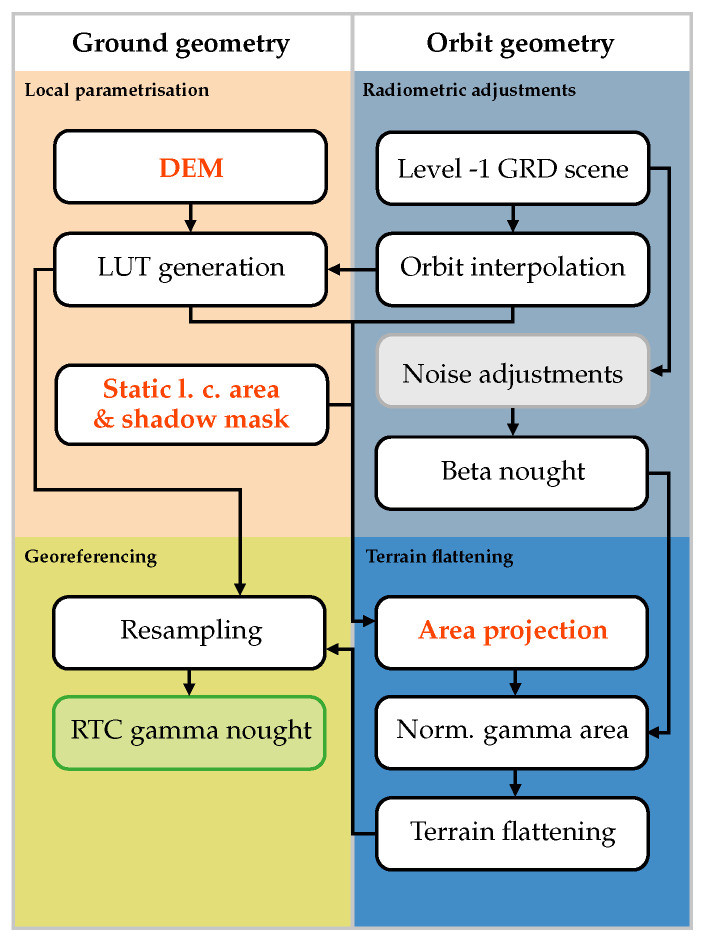
Improved Sentinel-1 RTC gamma nought pre-processing workflow. Modifications to the original workflow in Figure 1 are highlighted in red. Greyish boxes indicate common, but not mandatory steps.

**Figure 7 sensors-23-06072-f007:**
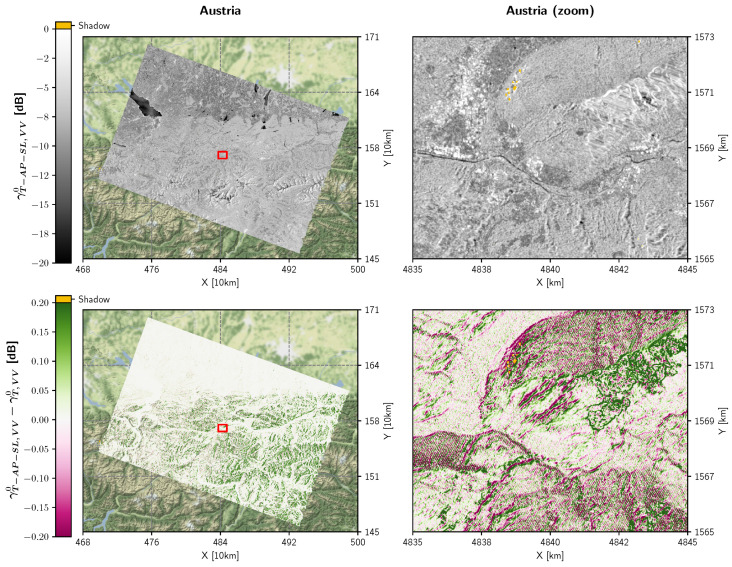
$\gamma_{T-\mathrm{AP}-\mathrm{SL},\;\mathrm{VV}}^{0}$ as an output of the improved Sentinel-1 RTC gamma nought pre-processing workflow shown in Figure 6 for the Austrian scene (top). The second row shows the difference between $\gamma_{T-\mathrm{AP}-\mathrm{SL},\;\mathrm{VV}}^{0}$ and the result $\gamma_{T,\;\mathrm{VV}}^{0}$ of the original workflow in Figure 1. Both versions do not apply oversampling. The red boxes on the left side indicate the extent of the zoom-in view on the right side.

**Table 1 sensors-23-06072-t001:** Summary of the benchmarking experiments for all different setups of Sentinel-1’s $\gamma_{T}^{0}$ pre-processing chain. Included additionally: the best setup of the Sentinel-1 GTC $\sigma_{E}^{0}$ pre-processing workflow taken from [29].

	Workflow	Ov. fac.	Scene Prep.	Aux. Data Prep.	RTC	Data Export	Total	w.r.t. Wizsard (Base)
**Norway**	wizsard (base)	1	42 s	10 s	11 min 56 s	9 s	12 min 57 s	-
	wizsard (base)	2	43 s	31 s	32 min 45 s	9 s	34 min 8 s	-
	wizsard (static $A_{\gamma }$)	1	42 s	13 s	10 min 45 s	9 s	11 min 49 s	−9%
	wizsard (static $A_{\gamma }$)	2	42 s	46 s	26 min 50 s	9 s	28 min 27 s	−17%
	wizsard (single LUT)	1	42 s	13 s	6 min 26 s	10 s	7 min 31 s	−42%
	wizsard (single LUT)	2	44 s	45 s	23 min 5 s	15 s	24 min 49 s	−27%
	wizsard (RTC-AP)	1	42 s	13 s	27 min 35 s	10 s	28 min 40 s	+121%
	wizsard (RTC-AP)	2	43 s	52 s	106 min 17 s	14 s	108 min 6 s	+217%
	SNAP 8	1	-	-	-	-	661 min 38 s	+5009%
	wizsard ($\sigma_{E}^{0}$)	1	-	-	-	-	5 min 52 s	−55%
**Austria**	wizsard (base)	1	44 s	6 s	6 min 10 s	4 s	7 min 4 s	-
	wizsard (base)	2	44 s	15 s	16 min 28 s	4 s	17 min 31 s	-
	wizsard (static $A_{\gamma }$)	1	44 s	7 s	5 min 27 s	5 s	6 min 23 s	−10%
	wizsard (static $A_{\gamma }$)	2	44 s	23 s	13 min 50 s	4 s	15 min 1 s	−14%
	wizsard (single LUT)	1	44 s	7 s	3 min 20 s	4 s	4 min 15 s	−40%
	wizsard (single LUT)	2	44 s	23 s	11 min 41 s	5 s	12 min 53 s	−26%
	wizsard (RTC-AP)	1	44 s	7 s	15 min 10 s	5 s	16 min 6 s	+128%
	wizsard (RTC-AP)	2	44 s	24 s	54 min 35 s	4 s	55 min 47 s	+218%
	SNAP 8	1	-	-	-	-	257 min 44 s	+3556%
	wizsard ($\sigma_{E}^{0}$)	1	-	-	-	-	3 min 17 s	−53%
**Benin**	wizsard (base)	1	43 s	4 s	4 min 24 s	3 s	5 min 14 s	-
	wizsard (base)	2	43 s	11 s	11 min 42 s	3 s	12 min 39 s	-
	wizsard (static $A_{\gamma }$)	1	43 s	5 s	3 min 55 s	3 s	4 min 46 s	−9%
	wizsard (static $A_{\gamma }$)	2	43 s	16 s	9 min 48 s	3 s	10 min 50 s	−14%
	wizsard (single LUT)	1	42 s	5 s	2 min 22 s	3 s	3 min 12 s	−39%
	wizsard (single LUT)	2	43 s	16 s	8 min 13 s	3 s	9 min 15 s	−27%
	wizsard (RTC-AP)	1	43 s	5 s	10 min 24 s	3 s	11 min 15 s	+115%
	wizsard (RTC-AP)	2	43 s	16 s	38 min 18 s	3 s	39 min 20 s	+211%
	SNAP 8	1	-	-	-	-	63 min 4 s	+1105%
	wizsard ($\sigma_{E}^{0}$)	1	-	-	-	-	2 min 29 s	−53%

## Data Availability

The data presented in this study are available on request from the corresponding author.

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
