# Peer review of "Utilising Sentinel-1’s Orbital Stability for Efficient Pre-Processing of Radiometric Terrain Corrected Gamma Nought Backscatter"

_sensors, 2023, doi:10.3390/s23136072_

Round 1
Reviewer 1 Report
Comments and Suggestions for Authors
Pag. 2 line: 80-83, if this is the objective of this work, it is suggested to improve the wording and specify what the objective of the work is.
Pag 3, line: 89-94 Delete this paragraph. It is an article, not a book.
Page 3, line 97-105. This paragraph requires improving the wording. As part of the Materials and Methods, describe the how, not the why.
Page 3, line 108: “We have graphically recycled the RTC gamma nought…..”
When writing a scientific article, it should be written in the third person (eg, it was done...).
Pag 4, line: 146, 179 and in various parts of the manuscript the wording is in the first person. Proofread and redact in third person throughout in manuscript.
Page 10, Fig 5. Improve the content and quality of the images. If applicable, delete them, as they do not provide significant information.
Page 15, line 523-534. Improve the writing of the conclusions. IDEM, write in the third person.
Pages 16-17, line 546-559. Delete this information. Abbreviations are cited in the body of the document only once between parentheses. The appendices are not typical of a manuscript or a scientific article. Eliminate.

The Quality of English is Ok
Reviewer 2 Report
This paper discusses an innovative approach to the pre-processing of Radiometric Terrain Corrected (RTC) gamma nought backscatter, a type of Synthetic Aperture Radar (SAR) data analysis. In this study, the authors propose a novel solution to reduce the computational requirements by utilizing the high orbital stability of Sentinel-1.
I have a few suggestions/queries below.
#1. The existing introduction of the paper lacks an adequate review of previous studies. The paper would benefit from a dedicated "Related Works" or "Literature Review" section where the authors could provide a thorough review of recent and relevant research in the field.
#2. The authors did an excellent job of demonstrating the advantages of their proposed method. However, a more detailed discussion of the potential limitations and challenges of their approach would be beneficial.
#3. The authors assert that their findings are relevant for all SAR missions with high spatiotemporal coverage and persistent orbital stability. However, they could consider discussing these implications in more detail. This could include discussing potential future applications of their method and its impact on other fields.
#4. Although the authors intentionally omitted oversampling in the final experiment to highlight the positive impacts of the RTC-AP method, it would be beneficial to also include a discussion on how oversampling could potentially affect the results. Would the differences in backscatter be more or less pronounced if oversampling were applied?
#5. The paper notes that in some instances, backscatter differences exceeded Sentinel-1’s relative radiometric accuracy of 0.1 dB. The authors should explain these instances in more detail. Why did this occur, and how does it affect the overall results and conclusions?
#6. The paper is generally well-written, but there are places where the language could be made clearer.
Minor improvements required
Reviewer 3 Report
written with care. however I am not familiar with the development of equation 6. or the derivative of equation 5 input to the error propagation formula. Perhaps you could make that transition clearer. I even looked at reference [27] for clarity but was unable to determine the clarity.
